# Label-Wise Graph Convolutional Network for Heterophilic Graphs

**Enyan Dai, Shijie Zhou, Zhimen Guo, Suhang Wang**

The Pennsylvania State University

{emd5759,smz5479,zhimeng,szw494}@psu.edu

## Abstract

Graph Neural Networks (GNNs) have achieved remarkable performance in modeling graphs for various applications. However, most existing GNNs assume the graphs exhibit strong homophily in node labels, i.e., nodes with similar labels are connected in the graphs. They fail to generalize to heterophilic graphs where linked nodes may have dissimilar labels and attributes. Therefore, we investigate a novel framework that performs well on graphs with either homophily or heterophily. Specifically, we propose a label-wise message passing mechanism to avoid the negative effects caused by aggregating dissimilar node representations and preserve the heterophilic contexts for representation learning. We further propose a bi-level optimization method to automatically select the model for graphs with homophily/heterophily. Theoretical analysis and extensive experiments demonstrate the effectiveness of our proposed framework (https://github.com/EnyanDai/LWGCN) for node classification on both homophilic and heterophilic graphs.

## 1 Introduction

Graph-structured data is very pervasive in the real-world such as knowledge graphs, traffic networks, and social networks. Therefore, it is important to model graphs for downstream tasks such as traffic prediction [45], recommendation system [19] and drug generation [3]. To capture the topology information in graphs, Graph Neural Networks (GNNs) [41] adopt a message-passing mechanism which learns a node's representation by iteratively aggregating the representations of neighbors. This can enrich the node features and preserve local topology for various downstream tasks.

Despite the great success of GNNs in modeling graphs, there is a concern in processing heterophilic graphs where edges often link nodes dissimilar in attributes or labels. Specifically, existing works [50, 9] find that GNNs could fail to generalize to graphs with heterophily due to their implicit/explicit homophily assumption. For example, Graph Convolutional Network (GCNs) is even outperformed by MLP that ignores the graph structure on heterophilic website datasets [50]. However, a recent work [32] argues that homophily assumption is not a necessity for GNNs. They show that GCN can work well on dense heterophilic graphs whose neighborhood patterns of different classes are distinguishable. But their analysis and conclusion is limited to the heterophilic graphs under strict conditions, and fails to show the relation between heterophily levels and performance of GNNs. Thus, in Sec. 3, we conduct thoroughly theoretical and empirical analysis on GCN to investigate the impacts of heterophily levels, which cover all the aforementioned observations. As the Theorem 1 and Fig. 2 show, the performance of GCN will firstly decrease then increase with the increment of heterophily levels. And the aggregation in GCN could even lead to non-discriminative representations under certain conditions.

Though heterophilic graphs challenge existing GNNs, the heterophilic neighborhood context itself provides useful information [32, 6]. Generally, two nodes of the same class tend to have similar heterophilic neighborhood contexts; while two nodes of different classes are more likely to have different heterophilic neighborhood contexts, which is verified in Appendix 3.3. Thus, a heterophilic

E. Dai et al., Label-Wise Graph Convolutional Network for Heterophilic Graphs. *Proceedings of the First Learning on Graphs Conference (LoG 2022)*, PMLR 180, Virtual Event, December 9–12, 2022.

context-preserving mechanism can lead to more discriminative representations. One promising way to preserve the heterophilic context is to conduct label-wise aggregation, i.e., separately aggregate neighbors in each class. In this way, we can summarize the heterophilic neighbors belonging to each class to an embedding to preserve the local context information for representation learning. As shown in the example in Fig. 3, for node $v_A$, with label-wise aggregation, $v_A$ will be represented as $[1.0, 5.5, 2.0, \text{non-existence}]$, in the order of $v_A$'s attribute, blue, green, and orange neighbors, respectively. Compared with $v_B$, $v_A$'s representations of central node and neighborhood context differ significantly with $v_B$. While for the aggregation in GCN, the obtained representations are rather similar for two nodes. In other words, we obtain more discriminative features on heterophilic graphs with label-wise aggregation, which is also verified by our analysis in Theorem 2. Though promising, there is no existing work on exploring label-wise message passing to address the challenge of heterophilic graphs.

Therefore, in this paper, we investigate novel label-wise aggregation for graph convolution to facilitate the node classification on heterophilic graphs. In essence, we are faced with two challenges: (**i**) the label-wise aggregation needs the label of each node; while for node classification, we are only given a small set of labeled nodes. How to adopt label-wise graph convolution on sparsely labeled heterophilic graphs to facilitate node classification? (**ii**) In practice, the homophily levels of the given graphs can be various and are often unknown. For homophily graphs, the label-wise graph convolution might not work as well as previous GNNs embedded with homophily assumption. How to ensure the performance on both heterophilic and homophilic graphs? In an attempt to address these challenges, we propose a novel framework L̲abel-W̲ise G̲C̲N̲ (LW-GCN). LW-GCN adopts a pseudo label predictor to predict pseudo labels and designs a novel label-wise message passing to preserve the heterophilic contexts with pseudo labels. To handle both heterophilic and homophilic graphs, apart from label-wise message passing GNN, LW-GCN also utilizes a GNN for homophilic graphs, and adopts bi-level optimization on the validation data to automatically select the better model for the given graph. The main contributions are:

- We theoretically show impacts of heterophily levels to GCN and demonstrate the potential limitations of GCN in learning on heterophilic graphs;

- We design a label-wise graph convolution to preserve the local context in heterophilic graphs, which is also proven by our theoretical and empirical analysis;

- We propose a novel framework LW-GCN, which deploys a pseudo label predictor and an automatic model selection module to achieve label-wise aggregation on sparsely labeled graphs and ensure the performance on both heterophilic and homophilic graphs; and

- Extensive experiments on real-world graphs with heterophily and homophily are conducted to demonstrate the effectiveness of LW-GCN.

## 2   Related Work

Graph neural networks (GNNs) have shown great success for various applications such as social networks [19, 12, 47], financial transaction networks [39, 17] and traffic networks [45, 46]. Based on the definition of the graph convolution, GNNs can be categorized into two categories, i.e., spectral-based [4, 16, 25, 27] and spatial-based [38, 42, 1]. Spectral-based GNN models are defined according to spectral graph theory. Bruna *et al.* [4] firstly generalize convolution operation to graph-structured data from spectral domain. GCN [25] simplifies the graph convolution by first-order approximation. For spatial-based graph convolution, it aggregates the information of the neighbors nodes [33, 19, 7]. Recently, to learn better node representations, deep graph neural networks [8, 26, 28] and self-supervised learning methods [37, 24, 44, 48] have been investigated. Moreover, explainable graph neural networks [13, 43, 14] and robust GNNs [10, 22, 11, 15] are also studied to address the problem of lacking trustworthiness in GNNs.

However, the aforementioned methods are generally designed based on the homophily assumption of the graph. Low homophily level in some real-word graphs can largely degrade their performance [50]. Some efforts [34, 2, 21, 50, 49, 9, 20, 30, 29] have been taken to address the problem of heterophilic graphs. For example, H2GCN [50] investigates three key designs for GNNs on heterophilic graphs. SimP-GCN [21] adopts a node similarity preserving mechanism to handle graphs with heterophiliy. FAGCN [2] adaptively aggregates low-frequency and high-frequency signals from neighbors to learn representations for graphs with heterophily. GPR-GNN [9] proposes a generalized PageRank GNN architecture that can learn positive/negative weights for the representations after different steps of

propagation to mitigate the graph heterophily issue. Recently, BM-GCN [20] proposes to utilize pseudo labels in the convolutional operation. Specifically, the pseudo labels are used to obtain a block similarity matrix to re-weight the edges in heterophilic graphs. Then, node pairs belonging to different label combinations could have different information exchange. Our LW-GCN is inherently different from these methods: (i) We propose a novel label-wise graph convolution to better capture the neighbors' information in heterophilic graphs; and (ii) Automatic model selection is deployed to achieve state-of-the-art performance on both homophilic and heterophilic graphs.

## 3 Preliminaries

In this section, we first present the notations and definition followed by the introduction of the GCN's design. We then conduct the theoretical analysis to investigate the impacts of heterophily to GCN.

### 3.1 Notations and Definition

Let $\mathcal{G} = (\mathcal{V}, \mathcal{E}, \mathbf{X})$ be an attributed graph, where $\mathcal{V} = \{v_1, ..., v_N\}$ is the set of $N$ nodes, $\mathcal{E} \subseteq \mathcal{V} \times \mathcal{V}$ is the set of edges, and $\mathbf{X} = \{\mathbf{x}_1, ..., \mathbf{x}_N\}$ is the set of node attributes. $\mathbf{A} \in \mathbb{R}^{N \times N}$ represents the adjacency matrix of the graph $\mathcal{G}$, where $\mathbf{A}_{ij} = 1$ indicates an edge between nodes $v_i$ and $v_j$; otherwise, $\mathbf{A}_{ij} = 0$. In the node classification task, each node belongs to one of $C$ classes. We use $y_i$ to denote label of node $v_i$. Graphs can be split into homophilic and heterophilic graphs based on how likely edges link nodes in the same class. The homophily level is measured by the homophily ratio:

**Definition 1 (Homophily Ratio)** *It is the fraction of edges in a graph that connect nodes of the same class. The homophily ratio $h$ is calculated as $h = \frac{|\{(v_i, v_j) \in \mathcal{E}: y_i = y_j\}|}{|\mathcal{E}|}$.*

When the homophily ratio is small, most of the edges will link nodes from different classes, which indicates a heterophilic graph. In homophilic graphs, connected nodes are more likely to belong to the same class, which will lead to a homophily ratio close to 1.

### 3.2 How does the Heterophily Affect the GCN?

GCN [25] is one of the most widely used graph neural networks. The operation in each layer of GCN can be written as:

$$\mathbf{H}^{(k+1)} = \sigma(\tilde{\mathbf{A}}\mathbf{H}^{(k)}\mathbf{W}^{(k)}), \tag{1}$$

where $\mathbf{H}^{(k)}$ is the node representation matrix of the output of the $k$-th layer and $\tilde{\mathbf{A}}$ is the normalized adjacency matrix. Generally, the symmetric normalized form $\mathbf{D}^{-\frac{1}{2}}\mathbf{A}\mathbf{D}^{-\frac{1}{2}}$ or $\mathbf{D}^{-1}\mathbf{A}$ is used as $\tilde{\mathbf{A}}$, where $\mathbf{D}$ is a diagonal matrix with $\mathbf{D}_{ii} = \sum_i \mathbf{A}_{ij}$. The adjacency matrix can be augmented with a self-loop. $\sigma$ is an activation function such as ReLU. In a single layer of GCN, the process can be split into two steps. First, GCN layer averages the neighbor features with $\mathbf{Z} = \tilde{\mathbf{A}}\mathbf{X}$. Then, a non-linear transformation $\sigma(\mathbf{Z}\mathbf{W})$ is applied to obtain intermediate features or final predictions. The step of averaging the neighbor features can benefit the node classification when the neighbors have similar features. However, for heterophilic graphs, mixing neighbors that possess different features may result in poor representations for node classification. This could be justified by the following theorem, which thoroughly analyzes the impacts of the heterophily level to the linear separability of the representations after one step aggregation in GCN.

**Assumptions.** We first discuss the assumptions of the heterophilic graphs: (**i**) Following previous works [50], the graph $\mathcal{G}$ is considered as a $d$-regular graph, i.e., each node has $d$ neighbors; For each node $v$, the label distribution of its neighbor node $u \in \mathcal{N}(v)$ follows $P(y_u = y_v|y_v) = h, P(y_u = y|y_v) = \frac{1-h}{C-1}, \forall y \neq y_v$. (**ii**) For nodes in different classes, their heterophilic neighbors' features follow different distributions and dimensions of features are independent to each other. Specifically, let $\mathcal{N}_k(v)$ denote node $v$'s neighbors of class $k$. For two nodes $v$ and $s$ in classes $i$ and $j$ ($i \neq j$), the features of their heterophilic neighbors $\mathcal{N}_k(v)$ and $\mathcal{N}_k(s)$ in class $k \in \{1, ..., C\}$ follow two different normal distributions $N(\boldsymbol{\mu}_{ik}, \boldsymbol{\sigma}_{ik})$ and $N(\boldsymbol{\mu}_{jk}, \boldsymbol{\sigma}_{jk})$, where $\boldsymbol{\mu}_{ik}$ and $\boldsymbol{\mu}_{jk}$ represent the means, $\boldsymbol{\sigma}_{ik}$ and $\boldsymbol{\sigma}_{ik}$ denote the standard deviations. Intuitively, though nodes in $\mathcal{N}_k(v)$ and $\mathcal{N}_k(s)$ belong to the same class $k$, they are connected to nodes of different classes because of their different properties. For example, in the molecule, the atom in the same class will exhibit different features, when they are linked to different atoms. Therefore, this assumption is valid. And it is also verified by the empirical analysis on large real-world heterophilic graphs in Sec. 3.3.

Let $\boldsymbol{\sigma}_i = \sqrt{\frac{1}{C}\sum_{k=1}^{C}(\boldsymbol{\mu}_{ik} - \bar{\boldsymbol{\mu}}_i)\odot(\boldsymbol{\mu}_{ik} - \bar{\boldsymbol{\mu}}_i)}$, where $\bar{\boldsymbol{\mu}}_i = \frac{1}{C}\sum_{k=1}^{C}\boldsymbol{\mu}_{ik}$ and $\odot$ represents the element-wise product. We can have the following theorem.

**Theorem 1** *For an attributed graph $\mathcal{G} = (\mathcal{V}, \mathcal{E}, \mathbf{X})$ that follows the above assumptions in Sec. 3.2, if $|\boldsymbol{\mu}_{ii} - \boldsymbol{\mu}_{jj}| > |\boldsymbol{\mu}_{ik} - \boldsymbol{\mu}_{jk}|$ and $\boldsymbol{\sigma}_i > \boldsymbol{\sigma}_{ii}$, $\forall k \in \{1, \dots C\}$, as the decrease of homophily ratio $h$, the discriminability of representations obtained by the averaging process in GCN layer, i.e. $\mathbf{Z} = \mathbf{D}^{-1}\mathbf{A}\mathbf{X}$, will firstly decrease until $h = \frac{1}{C}$ then increase. When $h = \frac{1}{C}$ and $d < \frac{\boldsymbol{\sigma}_i^2}{|\boldsymbol{\mu}_{ik} - \boldsymbol{\mu}_{jk}|^2}$, the representations after averaging process will be nearly non-discriminative.*

The detailed proof can be found in Appendix C. The conditions in this theorem generally hold. Since the intra-class distance is often much smaller than inter-calss distance, $|\boldsymbol{\mu}_{ii} - \boldsymbol{\mu}_{jj}| > |\boldsymbol{\mu}_{ik} - \boldsymbol{\mu}_{jk}|$ is generally meet in real-world graphs. As for $\boldsymbol{\sigma}_i$, it computes the standard deviations of mean neighbor features in different classes. As a result, $\boldsymbol{\sigma}_i$ is usually much larger than the $\boldsymbol{\sigma}_{ii}$ and $|\boldsymbol{\mu}_{ik} - \boldsymbol{\mu}_{jk}|$. Therefore, the Theorem 1 generally holds for the real-world graphs. And we can observe from Theorem 1 that (i) heterophily level in a certain range will largely degrade the performance of GCN; (ii) GCN will be more negatively affected by the heterophilic graphs with lower node degrees. Though our analysis is based on GCN, it can be easily extended to GNNs that average neighbor representations in the aggregation (e.g. GraphSage [19], APPNP [26], and SGC [40]). For the extension of the analysis on more complex message-passing mechanism, we leave it as future work.

### 3.3 Empirical Analysis on Heterophilic Graphs

**Justification of Assumption (ii) in Sec.3.2.** Specifically, we aim to show (i) For nodes in the same class, features of their neighbors in the same class are similar; (ii) For nodes in different classes, features of their neighbors in the same class follow different distributions. Let $\mathcal{X}_{ik} = \{x_u : y_u = k, y_v = i, u \in \mathcal{N}(v), v \in \mathcal{V}\}$ be the set of neighbors which belong to class $k$ and are linked by the central node in class $i$. For neighbors in class $k$, we analyze the average similarity scores between $\mathcal{X}_{ik}$ and $\mathcal{X}_{jk}$ to investigate whether neighbors in class $k$ that are linked by center nodes in different classes follow different distributions. Specifically, the average similarity score between $\mathcal{X}_{ik}$ and $\mathcal{X}_{jk}$ is obtained by

$$s(\mathcal{X}_{ik}, \mathcal{X}_{jk}) = \frac{1}{|\mathcal{X}_{ik}| \times |\mathcal{X}_{jk}|} \sum_{v_i \in \mathcal{X}_{ik}} \sum_{v_j \in \mathcal{X}_{jk}} \frac{\mathbf{x}_i \cdot \mathbf{x}_j}{\|\mathbf{x}_i\|\|\mathbf{x}_j\|}, \tag{2}$$

where $\mathbf{x}_i$ and $\mathbf{x}_j$ are features of node $v_i \in \mathcal{X}_{ik}$ and $v_j \in \mathcal{X}_{jk}$, respectively. The results on Crocodile, Chameleon, and Squirrel for representative neighbor classes are presented in Fig. 1, where $(i, j)$-th element in the similarity matrix denotes the average node feature cosine similarity between $\mathcal{X}_{ik}$ and $\mathcal{X}_{jk}$. From this figure, we can observe that:

- For $\mathcal{X}_{ik}, \forall i \in 1, \dots, C$, its intra-group similarity score is very high. This proves that the heterophilic neighbors' features are similar when the nodes are in the same class.
- The similarity scores between $\mathcal{X}_{ik}$ and $\mathcal{X}_{ik}$ are very small when $i \neq j$. This indicates that for nodes in different classes their heterophilic neighbors belonging to the same class still differs a lot.

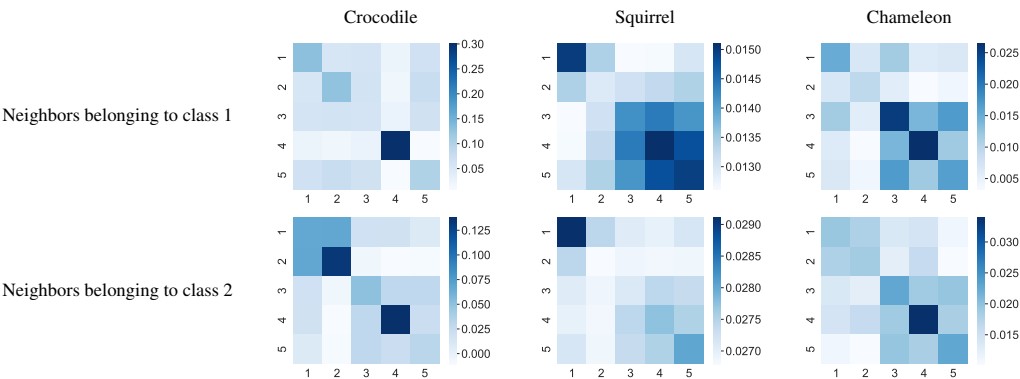

**Figure 1:** Similarity matrices of neighbors linked with centered nodes in different classes on Crocodile, Squirrel, and Chameleon.

With the above observations, Assumption (ii) is justified.

**Verification of Theorem 1.** To empirically verify the theoretical analysis of Theorem 1, we synthesize graphs with different homophily ratios and node degrees by deleting/adding edges in the crocodile graph following Appendix E.1. The results of GCN and GAT [38] on graphs with various node degrees are shown in Fig. 2. We can observe that (i) as the homophily ratio decreases the performance of GCN will keep decreasing until $h$ is around 0.2 ($h \approx \frac{1}{C}$), then the performance will start increase;(ii) when $h$ is around $\frac{1}{C}$, the performance can be very poor and even much worse than MLP on the graph with low node degrees. The observations are in consistent with our Theorem 1, which further demonstrates the general limitations of current GNN models in learning on graphs with heterophily. This trend

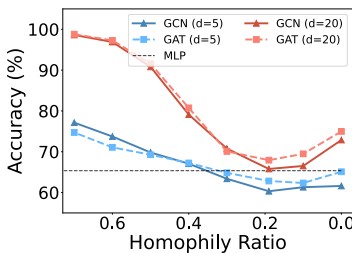

**Figure 2:** Impacts of the heterophily levels to GCN and GAT.

has also be reported in [50, 32, 31]. Moreover, theoretical analysis is conducted in [32] to prove the effectiveness of GCN on heterophilic graphs with discriminative neighborhoods. However, it can only explain the observation when $h < \frac{1}{C}$. By contrast, our theoretical analysis can well explain the whole trend of GCN performance w.r.t the homophily ratio. A similar conclusion is made with the theoretical analysis in [31], but node features are not incorporated and are replaced by label embedding vectors in their analysis.

### 3.4 Problem Definition

Based on the analysis above, we can infer that current GNNs are effective on graphs with high homophily; while they are challenged by the graphs with heterophily. In real world, we are usually given graphs with various homophily levels. In addition, the graphs are often sparsely labeled. And due to the lack of labels, the homophily ratio of the given graph is generally unknown. Thus, we aim to develop a framework that works for semi-supervised node classification on graphs with any homophily level. The problem is defined as:

**Problem 1** *Given an attributed graph $\mathcal{G} = (\mathcal{V}, \mathcal{E}, \mathbf{X})$ with a set of labels $\mathcal{Y}_L$ for node set $\mathcal{V}_L \subset \mathcal{V}$, the homophily ratio $h$ of $\mathcal{G}$ is unknown, we aim to learn a GNN which accurately predicts the labels of the unlabeled nodes, i.e., $f(\mathcal{G}, \mathcal{Y}_L) \rightarrow \hat{\mathcal{Y}}_U$, where $f$ is the function we aim to learn and $\hat{\mathcal{Y}}_U$ is the set of predicted labels for unlabeled nodes.*

## 4 Methodology

As the analysis in Sec. 3 shows, the aggregation process in GCN will mix the neighbors in various labels/distributions in heterophilic graphs, resulting in non-discriminative representations for local context. Based on this motivation, we propose to adopt label-wise aggregation in graph convolution, i.e., neighbors in the same class are separately aggregated, to preserve the heterophilic context. Next, we give the details of the label-wise aggregation along with the theoretical analysis that verifies its capability in obtaining distinguishable representations for heterophilic context. Then, we present how to apply label-wise graph convolution on sparsely labeled graphs and how to ensure performance on both heterophilic and homophilic graphs.

### 4.1 Label-Wise Graph Convolution

In heterophilic graphs, we observe that the heterophilic neighbor context itself provides useful information. Let $\mathcal{N}_k(v)$ denote node $v$'s neighbors of label class $k$. As shown in Appendix 3.3, for two nodes $u$ and $v$ of the same class, i.e., $y_u = y_v$, the features of nodes in $\mathcal{N}_k(u)$ are likely to be similar to that of nodes in $\mathcal{N}_k(v)$; while for nodes $u$ and $s$ with $y_u \neq y_s$, the features of nodes in $\mathcal{N}_k(u)$ are likely to be different from that in $\mathcal{N}_k(s)$. Therefore, for each node $v \in \mathcal{V}$, we propose to summarize the information of $\mathcal{N}_k(v)$ by label-wise aggregation to capture the useful heterophilic context. Let $\mathbf{a}_{v,k}$ be the aggregated representation of neighbors in class $k$, the process of obtaining representation for heterophilic context with the label-wise aggregation can be formally written as:

$$\mathbf{a}_{v,k} = \sum_{u \in \mathcal{N}_k(v)} \frac{1}{|\mathcal{N}_k(v)|} \mathbf{x}_u, \quad \mathbf{h}_v^c = \texttt{CONCAT}(\mathbf{a}_{v,1}, \dots, \mathbf{a}_{v,C}), \quad (3)$$

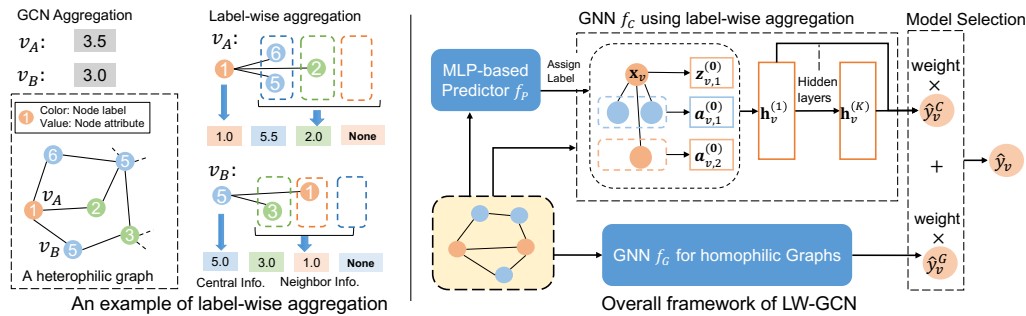

**Figure 3:** The illustration of label wise aggregation and overall framework of our LW-GCN.

where $C$ is the number of classes. $\mathbf{h}_v^c$ denotes the representation of the neighborhood context. As it is shown in Eq.(3), concatenation is applied to obtain representation of context to preserve the heterophilic context. When there is no neighbor of $v$ belonging to class $k$, zero embedding is assigned for class $k$. We then can augment the representation of the centered node with the context representation as the general design of GNNs. Specifically, we concatenate the context representation $\mathbf{h}_v^c$ and centered node representation $\mathbf{x}_v$ followed by the non-linear transformation:

$$\mathbf{h}_v = \sigma(\mathbf{W} \cdot \text{CONCAT}(\mathbf{x}_v, \mathbf{h}_v^c)), \tag{4}$$

where $\mathbf{W}$ denotes the learnable parameters in the label-wise graph convolution and $\sigma$ denotes the activation function such as ReLU.

In this section, we further prove the superiority of label-wise graph convolution in learning discrimative representations for heterophilic context by the following theorem.

**Theorem 2** *We consider an attributed graph $\mathcal{G} = (\mathcal{V}, \mathcal{E}, \mathbf{X})$ that follows the aforementioned assumptions in Sec. 3.2. If $|\boldsymbol{\mu}_{ik} - \boldsymbol{\mu}_{jk}| > \sqrt{\frac{C}{d}}\boldsymbol{\sigma}_{ik}, \forall k \in \{1, \ldots, C\}$, the heterophilic context representation $\mathbf{h}_v^c$ that is obtained by the label-wise aggregation with Eq.(3) will keep its discriminability regardless the value of homophily ratio $h$.*

The detailed proof is presented in Appendix D. The difference between the groups of neighbors is naturally larger than the intra-group variance. Since $\sqrt{\frac{C}{d}}$ is usually small (e.g. around 1.8 in the Texas graph), the condition $|\boldsymbol{\mu}_{ik} - \boldsymbol{\mu}_{jk}| > \sqrt{\frac{C}{d}}\boldsymbol{\sigma}_{ik}$ is generally satisfied in real-world scenarios. We also adopt the label-wise graph convolution on the synthetic graphs with different homophily ratios to empirically show its effectiveness. The results can be found in Appendix E.2.

## 4.2 LW-GCN: A Unified Framework for Graphs with Homophily or Heterophily

Though the analysis in Sec.3.1 proves the effectiveness of label-wise graph convolution in processing graphs with heterophily, there are still two major challenges for semi-supervised node classification on graphs with any heterophily levels: (i) how to conduct label-wise graph convolution on heterophilic graphs with a small number of labeled nodes; and (ii) how to make it work for both heterophilic and homophilic graphs. To address these challenges, we propose a novel framework LW-GCN, which is illustrated in Fig. 3. LW-GCN is composed of an MLP-based pseudo label predictor $f_P$, a GNN $f_C$ using label-wise graph convolution, a GNN $f_G$ for homophilic graph, and an automatic model selection module. The predictor $f_P$ takes the node attributes as input to give pseudo labels. $f_C$ utilizes the estimated pseudo labels from $f_P$ to conduct label-wise graph convolution on $\mathcal{G}$ for node classification. To ensure the performance on graphs with any homophily level, LW-GCN also trains $f_G$, i.e., a GNN for homophilic graphs, and can automatically select the model for graphs with unknown homophily ratios. For the model selection module, a bi-level optimization on validation set is applied to learn the weights for model selection. Next,we give the details of each component.

### 4.2.1 Pseudo Label Prediction.

In label-wise graph convolution, neighbors in different classes are separately aggregated to update node representations. However, only a small number of nodes are provided with labels. Thus, a

pseudo label predictor $f_P$ is deployed to estimate labels for label-wise aggregation. Specifically, a MLP is utilized to obtain pseudo label of node $v$ as $\hat{\mathbf{y}}_v^P = \text{MLP}(\mathbf{x}_v)$, where $\mathbf{x}_v$ is the attributes of node $v$. Note that, we use MLP as the predictor because message passing of the GNNs may lead to poor predictions on heterophilic graphs. The loss function for training $f_P$ is:

$$\min_{\theta_P} \mathcal{L}_P = \frac{1}{|\mathcal{V}_{train}|} \sum_{v \in \mathcal{V}_{train}} l(\hat{y}_v^P, y_v), \tag{5}$$

where $\mathcal{V}_{train}$ is the set of labeled nodes in the training set, $y_v$ denote the true label of node $v$, $\theta_P$ represents the parameters of the predictor $f_P$, and $l(\cdot)$ is the cross entropy loss.

### 4.2.2 Architecture of LW-GCN for Heterophilic Graphs.

With $f_P$, we can get pseudo labels $\hat{\mathcal{Y}}_U^P$ for unlabeled nodes $\mathcal{V}_U = \mathcal{V} \backslash \mathcal{V}_L$. Combining it with the provided $\mathcal{Y}_L$, we have labels $\mathcal{Y}^P \in (\hat{\mathcal{Y}}_U^P \cup \mathcal{Y}_L)$ necessary for label-wise aggregation in Eq.(3). Then, node representations can be updated with the heterophilic context by Eq.(4). Multiple layers of label-wise graph convolution can be applied to incorporate more hops of neighbors in representation learning. The process of one layer label-wise graph convolution with pseudo labels can be rewritten as:

$$\mathbf{a}_{v,k}^{(l)} = \sum_{u \in \mathcal{N}_k^P(v)} \frac{1}{|\mathcal{N}_k^P(v)|} \mathbf{h}_u^{(l)}, \quad \mathbf{h}_v^{l+1} = \sigma\Big(\mathbf{W}^{(l)} \cdot \text{CONCAT}(\mathbf{h}_v^{(l)}, a_{v,1}^{(l)}, \dots, a_{v,C}^{(l)})\Big), \tag{6}$$

where $\mathcal{N}_k^P(v) = \{u : (v, u) \in \mathcal{E} \wedge \hat{y}_u^P = k\}$ stands for node $v$'s neighbors with estimated label $k$. $\mathbf{h}_v^{(l)}$ is the representation of node $v$ at the $l$-th layer label-wise graph convolution with $\mathbf{h}_v^{(0)} = \mathbf{x}_v$. In heterophilic graphs, different hops of neighbors may exhibit different distributions which can provide useful information for node classification. Therefore, the final node prediction can be conducted by combining the intermediate representations of the model with $K$ layers:

$$\hat{\mathbf{y}}_v^C = \text{softmax}\Big(\mathbf{W}_C \cdot \text{COMBINE}(\mathbf{h}_v^{(1)}, \dots, \mathbf{h}_v^{(K)})\Big), \tag{7}$$

where $\mathbf{W}_C$ is a learnable weight matrix, $\hat{\mathbf{y}}_v^C$ is predicted label probabilities of node $v$. Various operations such as max-pooling and concatenation [42] can be applied as the COMBINE function.

### 4.2.3 Automatic Model Selection

In heterophilic graphs, the homophily ratio is very small and even can be around 0.2 [34]. With a reasonable pseudo label predictor, the label-wise aggregation with pseudo labels will mix much less noise than the general GNN aggregation. In contrast, for homophilic graphs such as citation networks, their homophily ratios are close to 1. In this situation, directly aggregating all the neighbors may introduce less noise in representations than aggregating label-wisely as the pseudo-labels contain noises. Therefore, it is necessary to determine whether to apply the label-wise graph convolution or the state-of-the-art GNN for homophilic graphs. One straightforward way is to select the model based on the homophily ratio. However, graphs are generally sparsely labeled which makes it difficult to estimate the real homophily ratio. To address this problem, we propose to utilize the validation set to automatically select the model.

In the model selection module, we combine predictions of the label-wise aggregation model for heterophilic graphs and traditional GNN models for homophilic graphs. Predictions from the GNN $f_G$ for homophilic graphs are given by $\hat{\mathcal{Y}}^G = \text{GNN}(\mathbf{A}, \mathbf{X})$, where the GNN is flexible to various models for homophilic graphs. Here, we select GCNII [8] which achieves state-of-the-art results on homophilic graphs. The model selection can be achieved by assigning higher weight to the corresponding model prediction. The combined prediction is given as:

$$\hat{\mathbf{y}}_v = \frac{\exp(\phi_1)}{\sum_{i=1}^{2} \exp(\phi_i)} \hat{\mathbf{y}}_v^C + \frac{\exp(\phi_2)}{\sum_{i=1}^{2} \exp(\phi_i)} \hat{\mathbf{y}}_v^G, \tag{8}$$

where $\hat{\mathbf{y}}_v^G \in \hat{\mathcal{Y}}^G$ is the prediction of node $v$ from $f_G$. $\phi_1$ and $\phi_2$ are the learnable weights to control the contributions of two models in final prediction. $\phi_1$ and $\phi_2$ can be obtained by finding the values that lead to good performance on validation set. More specifically, this goal can be formulated as the following bi-level optimization problem:

$$\min_{\phi_1, \phi_2} \mathcal{L}_{val}(\theta_C^*(\phi_1, \phi_2), \theta_G^*(\phi_1, \phi_2), \phi_1, \phi_2) \quad s.t. \ \theta_C^*, \theta_G^* = \arg\min_{\theta_C, \theta_G} \mathcal{L}_{train}(\theta_C, \theta_G, \phi_1, \phi_2) \tag{9}$$

where $\mathcal{L}_{val}$ and $\mathcal{L}_{train}$ are the average cross entropy loss of the combined predictions $\{\hat{y}_v : v \in \mathcal{V}_{val}\}$ and $\{\hat{y}_v : v \in \mathcal{V}_{train}\}$ on validation set and training set, respectively.

### 4.3 An Optimization Algorithm of LW-GCN

Computing the gradients for $\phi_1$ and $\phi_2$ is expensive in both computational cost and memory. To alleviate this issue, we use an alternating optimization schema to iteratively update the model parameters and the model selection weights.

**Updating Lower Level $\theta_C$ and $\theta_G$.** Instead of calculating $\theta_C^*$ and $\theta_G^*$ per outer iteration, we fix $\phi_1$ and $\phi_2$ and update the mode parameters $\theta_G$ and $\theta_C$ for $T$ steps by:

$$\theta_C^{t+1} = \theta_C^t - \alpha_C \nabla_{\theta_C} \mathcal{L}_{train}(\theta_C^t, \theta_G^t, \phi_1, \phi_2), \quad \theta_G^{t+1} = \theta_G^t - \alpha_G \nabla_{\theta_G} \mathcal{L}_{train}(\theta_C^t, \theta_G^t, \phi_1, \phi_2), \tag{10}$$

where $\theta_C^t$ and $\theta_G^t$ are model parameters after updating $t$ steps. $\alpha_C$ and $\alpha_G$ are the learning rates for $\theta_C$ and $\theta_G$.

**Updating Upper Level $\phi_1$ and $\phi_2$.** Here, we use the updated model parameters $\theta_C^T$ and $\theta_G^T$ to approximate $\theta_C^*$ and $\theta_G^*$. Moreover, to further speed up the optimization, we apply first-order approximation [18] to compute the gradients of $\phi_1$ and $\phi_2$:

$$\phi_1^{k+1} = \phi_1^k - \alpha_\phi \nabla_{\phi_1} \mathcal{L}_{val}(\bar{\theta}_C^T, \bar{\theta}_G^T, \phi_1^k, \phi_2^k), \quad \phi_2^{k+1} = \phi_2^k - \alpha_\phi \nabla_{\phi_2} \mathcal{L}_{val}(\bar{\theta}_C^T, \bar{\theta}_G^T, \phi_1^k, \phi_2^k), \tag{11}$$

where $\bar{\theta}_C^T$ and $\bar{\theta}_G^T$ means stopping the gradient. $\alpha_\phi$ is the learning rate for $\phi_1$ and $\phi_2$.

More details of the training algorithm are in Appendix A.

## 5 Experiments

In this section, we conduct experiments to demonstrate the effectiveness of LW-GCN. In particular, we aim to answer the following research questions:

- **RQ1** Is our LW-GCN effective in node classification on both homophilic and heterophilic graphs?
- **RQ2** How do the quality of pseudo labels and the automatic model selection affect LW-GCN?
- **RQ3** Can label-wise aggregation learn representations that well capture information for prediction?

### 5.1 Experimental Settings

**Datasets.** For homophilic graphs, we choose the widely used benchmark datasets, Cora, Citeseer, and Pubmed [25]. The dataset splits of homophilic graphs are the same as the cited paper. As for heterophilic graphs, we use three webpage datasets Texas, Cornell, and Wisconsin [34], and three subgraphs of wiki, i.e., Squirrel, Chameleon, and Crocodile [36]. Following [50], 10 dataset splits are used in each heterophilic graph for evaluation. In addition, we also use a large scale heterophilc citation network, i.e., arxiv-year [30]. 5 public splits of arxiv-year are used for evaluation. The statistics of the datasets are presented in Table 3 in the Appendix.

**Compared Methods.** We compare LW-GCN with state-of-the-art GNNs, which includes GCN [25], MixHop [24], SuperGAT [23], and GCNII [8]. We also compare with the following state-of-the-art models designed for heterophilic graphs: FAGCN [2], SimP-GCN [21], H2GCN [50], GRP-GNN [9], BM-GCN [20], ASGC [5], LINKX [30] and GloGNN++ [29]. In addition, the MLP are evaluated on the datasets for reference. The details of these compared methods can be found in Appendix B.2.

**Settings of LW-GCN.** For the label predictor $f_P$, we adopt a MLP with one-hidden layer. As for the $f_C$, we adopt two layers of label-wise message passing on all the datasets. More discussion about the impacts of the depth on LW-GCN is given in Sec. G. The other hyperparameters such as hidden dimension and weight decay are tuned based on the validation set. See Appendix B.1 for more details.

### 5.2 Node Classification Performance

To answer **RQ1**, we conduct experiments on both heterophilic graphs and homophilic graphs. The average accuracy and standard deviations on homophilic/heterophilic graphs are reported in Table 1. Additional results on Cornell and Citeseer datasets are presented in Appendix F. The model selection weight for label-wise aggregation GNN $f_C$ is shown along with the results of LW-GCN. Note that this

**Table 1:** Node classification results (Accuracy(%) $\pm$ Std.) on homophilic/heterophilic graphs.

| Dataset | Wisconsin | Texas | Chameleon | Squirrel | Crocodile | arxiv-year | Cora | Pubmed |
|---|---|---|---|---|---|---|---|---|
| Ave. Degree | 2.05 | 1.69 | 15.85 | 41.74 | 30.96 | 6.9 | 4.01 | 4.50 |
| Homo. Ratio | 0.20 | 0.11 | 0.24 | 0.22 | 0.25 | 0.22 | 0.81 | 0.8 |
| MLP | 83.5 $\pm$4.9 | 78.1 $\pm$6.0 | 48.0 $\pm$1.5 | 32.3 $\pm$1.8 | 65.8 $\pm$0.7 | 36.7 $\pm$0.2 | 58.6 $\pm$0.5 | 72.7 $\pm$0.4 |
| GCN | 53.1 $\pm$5.8 | 57.6 $\pm$5.9 | 63.5 $\pm$2.5 | 46.7 $\pm$1.5 | 66.7 $\pm$1.0 | 46.0 $\pm$0.3 | 81.6 $\pm$0.7 | 78.4 $\pm$1.1 |
| MixHop | 70.2 $\pm$4.8 | 60.6 $\pm$7.7 | 61.2 $\pm$2.2 | 44.1 $\pm$1.1 | 67.6 $\pm$1.3 | 46.1 $\pm$0.5 | 80.6 $\pm$0.2 | 78.9 $\pm$0.5 |
| SuperGAT | 53.7 $\pm$5.7 | 58.6 $\pm$7.7 | 59.4 $\pm$2.5 | 38.9 $\pm$1.5 | 62.6 $\pm$0.9 | 38.1 $\pm$0.1 | 82.7 $\pm$0.4 | 78.4 $\pm$0.5 |
| GCNII | 82.1 $\pm$3.9 | 68.6 $\pm$9.8 | 63.5 $\pm$2.5 | 49.4 $\pm$1.7 | 69.0 $\pm$0.7 | 47.2 $\pm$0.3 | 84.2 $\pm$0.5 | 80.2 $\pm$0.2 |
| FAGCN | 83.3 $\pm$3.7 | 79.5 $\pm$4.8 | 63.9 $\pm$2.2 | 43.3 $\pm$2.5 | 67.1 $\pm$0.9 | 40.6 $\pm$0.4 | 83.1 $\pm$0.6 | 78.8 $\pm$0.3 |
| SimP-GCN | 85.5 $\pm$4.7 | 80.5 $\pm$5.9 | 63.7 $\pm$2.3 | 42.8 $\pm$1.4 | 63.7 $\pm$2.3 | OOM | 82.8 $\pm$0.1 | 80.3 $\pm$0.2 |
| H2GCN | 84.7 $\pm$3.9 | 83.7 $\pm$6.0 | 54.2 $\pm$2.3 | 36.0 $\pm$1.1 | 66.7 $\pm$0.5 | 49.1 $\pm$0.1 | 81.6 $\pm$0.4 | 79.5 $\pm$0.2 |
| GPRGNN | 78.2 $\pm$4.4 | 77.0 $\pm$6.4 | 70.6 $\pm$2.1 | 50.8 $\pm$1.4 | 65.6 $\pm$0.9 | 45.1 $\pm$0.2 | 83.8 $\pm$0.6 | 79.9 $\pm$0.1 |
| BM-GCN | 77.6 $\pm$5.9 | 81.9 $\pm$5.4 | 69.4 $\pm$1.7 | 53.1 $\pm$1.8 | 64.3 $\pm$1.1 | OOM | 81.5 $\pm$0.5 | 77.9$\pm$0.4 |
| ASGC | 84.3 $\pm$2.6 | 85.9 $\pm$4.7 | 68.8 $\pm$1.6 | 54.5 $\pm$1.6 | 66.4 $\pm$0.7 | 39.2 $\pm$0.1 | 76.8 $\pm$0.2 | 74.4 $\pm$0.1 |
| LINKX | 75.5 $\pm$5.7 | 74.6 $\pm$8.4 | 68.4 $\pm$1.4 | 61.8 $\pm$1.8 | 79.4 $\pm$0.6 | **56.0 $\pm$1.3** | 64.7 $\pm$0.4 | 70.4 $\pm$0.7 |
| GloGNN++ | **88.0 $\pm$3.2** | 83.2 $\pm$4.3 | 71.2 $\pm$2.5 | 57.9 $\pm$2.0 | 78.4 $\pm$0.9 | 54.8 $\pm$0.3 | 66.7 $\pm$1.9 | 78.1 $\pm$0.2 |
| LW-GCN | 86.9 $\pm$2.2 | **86.2 $\pm$5.8** | **74.4 $\pm$1.4** | **62.6 $\pm$1.6** | **79.7 $\pm$0.4** | 55.8 $\pm$0.2 | **84.3 $\pm$0.3** | **80.4 $\pm$0.3** |
| Weight for $f_C$ | 0.981 | 0.960 | 0.986 | 0.987 | 0.999 | 0.942 | 0.001 | 0.006 |

model selection weight ranges from 0 to 1. When the weight is close to 1, the label-wise aggregation model is selected. When the weight for $f_C$ is close to 0, the GNN $f_G$ for homophilic graph is selected.

**Performance on Heterophilic Graphs.** We conduct experiments on 10 dataset splits on each heterophilic graph. From the results on heterophilic graphs, we can have following observations:

- MLP outperforms GCN and other GNNs for homophilic graphs by a large margin on Texas and Wisconsin; while GCN can achieve relatively good performance on dense heterophilic graphs such as Chameleon. This empirical result is consistent with our analysis in Theorem 1 that the heterophily will especially degrade the performance of GCN on graphs with low degrees.

- Though GCN and other GNNs designed for homophilic graphs can give relatively good performance on dense heterophilic graphs, our LW-GCN bring significant improvement by adopting label-wise aggregation. In addition, LW-GCN outperforms baselines on heterophilic graphs with low node degrees. This proves the superiority of label-wise aggregation in preserving heterophilic context.

- The model selection weight for $f_C$ is close to 1 for heterophilic graphs, which verifies that the proposed LW-GCN can correctly select the label-wise aggregation GNN $f_C$ for heterophilic graphs.

- Compared with SimP-GCN which also aims to preserve node features, our LW-GCN performs significantly better on heterophilic graphs. This is because SimP-GCN only focuses on the similarity of central node attributes. In contrast, our label-wise aggregation can preserve both the central node features and the heterophilic local context for node classification. LW-GCN also outperforms the other GNNs that adopt message-passing mechanism designed for heterophilic graphs by a large margin. This further demonstrates the effectiveness of label-wise aggregation.

**Performance on Homophilic Graphs.** The average results and standard deviations of 5 runs on homophilic graphs, i.e., Cora, Citeseer, and Pubmed, are also reported in Table 1 and Appendix F. From the results, we can observe that existing GNNs for heterophilic graphs generally perform worse than state-of-the-art GNNs on homophilic graphs such as GCNII. In contrast, LW-GCN achieves comparable results with the the best model on homophilic graphs. This is because LW-GCN combines the GNN using label-wise message passing and a state-of-the-art GNN for homophilic graph. And it can automatically select the right model for the given homophilic graph.

### 5.3 Ablation Study

To answer **RQ2**, we conduct ablation studies to understand the contributions of each component to LW-GCN. To investigate how the quality of pseudo labels can affect LW-GCN, we train a variant LW-GCN\P by replacing the MLP-based label predictor with a GCN model. To show the importance of the automatic model selection, we train a variant LW-GCN\G which removes the GNN for homophilic graphs and only uses label-wise aggregation GNN. Finally, we replace the GCNII backbone of $f_G$ to GCN, denoted as LW-GCN$_{GCN}$, to show LW-GCN is flexible to adopt various GNNs for $f_G$. Experiments are conducted on both homophilic and heterophilic graphs. The results are shown in Table 2. We can observe that:

**Table 2:** Ablation Study

| Dataset | MLP | GCN | GCNII | LW-GCN\P | LW-GCN\G | LW-GCN$_{GCN}$ | LW-GCN |
|---|---|---|---|---|---|---|---|
| Cora | 58.7 ±0.5 | 81.6 ±0.7 | 84.2 ±0.5 | 84.2 ±0.3 | 75.3 ±0.4 | 81.9 ±0.2 | **84.3** ±0.3 |
| Citeseer | 60.3 ±0.4 | 71.3 ±0.3 | 72.0 ±0.8 | 72.3 ±0.5 | 65.1 ±0.5 | 71.6 ±0.3 | **72.3** ±0.4 |
| Pubmed | 72.7 ±0.4 | 78.4 ±1.1 | 80.2 ±0.2 | 77.6 ±0.7 | 72.4 ±0.6 | 79.2 ±0.8 | **80.3** ±0.3 |
| Texas | 78.1 ±6.0 | 57.6 ±5.9 | 68.6 ±9.8 | 82.4 ±5.2 | 85.9 ±5.6 | 85.4 ±6.3 | **86.2** ±5.8 |
| Chameleon | 48.0 ±1.5 | 63.5 ±2.5 | 63.5 ±2.5 | **74.7** ±1.4 | 74.2 ±1.8 | 74.3 ±2.3 | 74.4 ±1.2 |
| Squirrel | 32.3 ±1.8 | 46.7 ±1.5 | 49.4 ±1.7 | 62.3 ±2.3 | 62.3 ±1.3 | 61.9 ±1.4 | **62.6** ±1.6 |

- On homophilic graphs, LW-GCN\P shows comparable results with LW-GCN, because GCNII will be selected given a homophilic graph. On the heterophilic graph Texas, the performance of LW-GCN\P is significantly worse than LW-GCN. This is because GNNs can produce poor pseudo labels on heterophilic graph, which degrades the label-wise message passing.

- LW-GCN\G performs much better than MLP. This shows label-wise graph convolution can capture structure information. However, LW-GCN\G performs worse than GCNII and LW-GCN on homophilic graphs, which indicates the necessity of combining GNN for homophilic graphs.

- LW-GCN$_{GCN}$ achieves comparable results with GCN on homophilic graphs. On heterophilic graphs, LW-GCN$_{GCN}$ performs similarly with LW-GCN. This shows the flexibility of LW-GCN in adopting traditional GNN models designed for homophilic graphs.

### 5.4 Analysis of Node Representations

To answer **RQ3**, we compare the representation similarity of intra-class and inter-class node pairs in Fig. 4. For both GCN and LW-GCN, representations in the last layer are used for analysis. we can observe that the representations of GCN are very similar for both intra-class pairs and inter-class pairs. This verifies that simply averaging the neighbors will lead to less discrimative representations . With label-wise aggregation, the similarity scores of intra-class pairs are significantly higher than inter-class node pairs. This demonstrates that the representations learned by label-wise aggregation can well preserve the nodes' features and their contextual information.

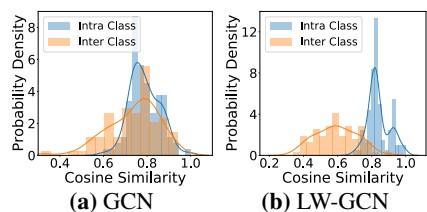

**(a)** GCN  **(b)** LW-GCN

**Figure 4:** Representation similarity distributions on Texas Graphs.

## 6 Conclusion and Future Work

In this paper, we analyze the impacts of the heterophily levels to GCN model and demonstrate its limitations. We develop a novel label-wise graph convolution to preserve the heterophilic neighbors' information. An automatic model selection module is applied to ensure the performance of the proposed framework on graphs with any homophily ratio. Theoretical and empirical analysis demonstrates the effectiveness of the label-wise aggregation. There are several interesting directions need further investigation. First, since better pseudo labels will benefit the label-wise message passing, it is promising to incorporate the predictions of LW-GCN in label-wise message passing. Second, in some applications such as link prediction, labels are not available. Therefore, we will investigate how to generate useful pseudo labels for label-wise aggregation for applications without labels.

## Acknowledgements

This material is based upon work supported by, or in part by, the National Science Foundation (NSF) under grant number IIS-1909702, the Army Research Office (ONR) under grant number W911NF21-1-0198, and Department of Homeland Security (DNS) CINA under grant number E205949D. The findings and conclusions in this paper do not necessarily reflect the view of the funding agencies.

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

---

**Algorithm 1** Training Algorithm of LW-GCN

---

**Input:** $\mathcal{G} = (\mathcal{V}, \mathcal{E}, X)$, $\mathcal{Y}_L$, $p$, $\alpha_C$, $\alpha_G$, $\alpha_\phi$ and $T$
**Output:** $f_P$, $f_C$, $f_G$, $\phi_1$ and $\phi_2$
 1: Train $f_P$ by optimizing Eq.(5) w.r.t $\theta_P$
 2: Obtain pseudo labels $\tilde{\mathcal{Y}}^P$ with $f_P$
 3: **repeat**
 4:    Get combined predictions of $f_C$ and $f_G$ on $\mathcal{V}_{val}$
 5:    Calculate the upper level loss $\mathcal{L}_{val}$
 6:    Update $\phi_1$ and $\phi_2$ according to Eq.(11)
 7:    **for** $t = 1$ to $T$ **do**
 8:       Obtain the lower level loss $\mathcal{L}_{train}$
 9:       Update $\theta_C$ and $\theta_G$ by Eq.(10)
10:    **end for**
11: **until** convergence

---

**Table 3:** The statistics of datasets.

| Dataset | Nodes | Edges | Classes | Hom. Ratio |
|---|---|---|---|---|
| Wisconsin | 251 | 515 | 5 | 0.20 |
| Texas | 183 | 309 | 5 | 0.11 |
| Cornell | 183 | 280 | 5 | 0.30 |
| Chameleon | 2,277 | 36,101 | 5 | 0.24 |
| Squirrel | 5,201 | 217,073 | 5 | 0.22 |
| Crocodile | 11,631 | 360,040 | 5 | 0.25 |
| arxiv-year | 169,343 | 1,166,243 | 5 | 0.22 |
| Cora | 2,708 | 5,429 | 6 | 0.81 |
| Citeseer | 3,327 | 4,732 | 7 | 0.74 |
| Pubmed | 19,717 | 44,338 | 3 | 0.8 |

## A   Training Algorithm of LW-GCN

The training algorithm of LW-GCN is shown in Algorithm 1. In line 1 and 2, we firstly train the $f_P$ to obtain the required pseudo labels for label-wise message passing. From line 4 to 6, we get the combined predictions from $f_C$ and $f_G$ and update the model selection weights with Eq.(11). From line 7 to 10, we update the model parameters $\theta_C$ and $\theta_G$ by minimizing $\mathcal{L}_{train}$ with Eq.(10). The updating of model selection weights and model parameters are conducted iteratively until convenience.

## B   Additional Details of Experimental Settings

### B.1   Implementation Details of LW-GCN

For experiments on each heterophilic graph, we report the results on the 10 public dataset splits. For homophily graphs, we run each experiment 5 times on the provided public dataset split. The hidden dimension of $f_P$ is fixed as 64 for all graphs. For the $f_C$ on Texas and Wisconsin, a linear layer is firstly applied to transform the features followed by the label-wise graph convolutional layer. As for the other graphs, the label-wise graph convolutional layer is directly applied to the node features. The hidden layer dimension and weight decay rate are tuned based on the validation set by grid search. Specifically, we vary the hidden dimension and weight decay in $\{32, 64, 128, 256\}$ and $\{0.05, 0.005, 0.0005, 0.00005\}$, respectively . As for the $f_G$ which deploys GCNII [8] as the backbone, the hyperparameter settings are the same as the cited paper. During the training phase, the learning rate is set as 0.01 for all the parameters and model selection weights. The inner iteration step $T$ is set as 1. Our machine uses an Intel i7-9700k CPU with 64GB RAM. A Nvidia 2080Ti GPU is used to run all the experiments.

### B.2   Implementation Details of Compared Methods

We adopt a two-layer MLP model on the datasets as baslines to show the effects of the graph structure and local context of the graphs. The hidden dimension is set the same as our LW-GCN. Apart

from MLP, we compare LW-GCN with the following representative and state-of-the-art GNNs that originally designed for graphs with homophily:

- **GCN** [25]: This is a popular spectral-based Graph Convolutional Network, which aggregates the neighbor information and the centered node by averaging their representations. We apply the official code in `https://github.com/tkipf/pygcn`.

- **MixHop** [1]: It adopts a graph convolutional layer with powers of the adjacency matrix. The official code in `https://github.com/samihaija/mixhop` is implemented for comparsion.

- **SuperGAT** [23]: This is a GAT model augmented by the self-supervision. In SuperGAT, apart from the classification loss on provided labels, a self-supervised learning task is deployed to further guide the learning of attention for better information propagation based on GAT [38]. The official code from the authors in `https://github.com/dongkwan-kim/SuperGAT` is used.

- **GCNII** [8]: Based on GCN, residual connection and identity mapping are applied in GCNII to have a deep GNN for better performance. The experiments are run with the official implementation in `https://github.com/chennnM/GCNII`.

We also compare LW-GCN with the following baseline GNN models for heterophilic graphs:

- **FAGCN** [2]: FAGCN adaptively aggregates low-frequency and high-frequency signals from neighbors to improve the performance on heterophilic graphs. The implementation from authors in `https://github.com/bdy9527/FAGCN` is applied in our experiments.

- **SimP-GCN** [21]: A feature similarity preserving aggregation is applied to facilitate the representation learning on graphs with homophily and heterophily. We utilize the official code in `https://github.com/ChandlerBang/SimP-GCN`.

- **H2GCN** [50]: H2GCN investigates the limitations of GCN on graphs with heterophily. And it accordingly adopts three key designs for node classification on heterophilic graphs. We conduct experiments with the official code from authors in `https://github.com/GemsLab/H2GCN`.

- **GPR-GNN** [9]: This method introduces a new Generalized PageRank (GPR) GNN to adaptively learn the GPR weights that combine the aggregated representations in different orders. The learned GPR weights can be either positive or negative, which allows the GPR-GNN handle both heterophilic and homophilic graphs. We adopt the official code from authors in `https://github.com/jianhao2016/GPRGNN`.

- **BM-GCN** [20]: This is one of the most recent methods designed for graphs with heterophily, which achieves state-of-the-art results on heterophilic graphs. A block-modeling is adopted to GCN to aggregate information from homophilic and heterophilic neighbors discrimatively. More specifically, the link between two nodes will be re-weighted based on the soft labels of two nodes and the block-similarity matrix. The training and evaluation process is based on the official code in `https://github.com/hedongxiao-tju/BM-GCN`.

- **ASGC** [5]: This method replaces the fixed feature propagation step of SGC [40] with an adaptive propagation, which can be effective for both homophilic graphs and heterophilic graphs. We use the official code released in `https://openreview.net/forum?id=jRrpiqxtrWm`.

- **LINKX** [30]: This methods separately embed the adjacency matrix and node features with multilayer perceptrions and simple transformations. We use the official code from authors in `https://github.com/CUAI/Non-Homophily-Large-Scale`.

- **GloGNN++** [29]: This method will learn a coefficient matrix to capture the correlations between nodes to aggregate information from global nodes in the graph. The values of the coefficient matrix can be signed and are derived from the optimization. In our experiments, we use the official code in `https://github.com/recklessronan/glognn`.

The model architecture and hyperparameters of the baselines are set according to the experimental settings provided by the authors for reproduction. For datasets that are not given reproduction details, the hyperparameters of baselines will be tuned based on the performance on validation set to make a fair comparison.

## C    Proof of Theorem 1

**Proof 1** *In this proof, we focus on nodes in class $i$ and class $j$, where $i \neq j$. Since dimensions of the node feature are independent to each other, without loss of generality, we consider one dimension of the feature and aggregated representation for node $v$, which is denoted as $x_v$ and $z_v$. For node $v$ in*

*class $i$, the aggregated representation $z_v$ in GCN layer is rewritten as:*

$$z_v = \sum_{u \in \mathcal{N}(v)} \frac{1}{|\mathcal{N}(v)|} x_u. \tag{12}$$

*With assumptions in Sec. 3.2, the expectation of aggregated representations of nodes in class $i$ can be written as:*

$$\mathbb{E}(z_v|y_v = i) = h \cdot \mu_{ii} + \frac{1-h}{C-1} \sum_{k=1, k \neq i}^{C} \mu_{ik}, \tag{13}$$

*Similarly, we can get the expectation of aggregated nodes representations in class $j$, i.e., $\mathbb{E}(z_v|y_v = j)$. Then, the difference between $\mathbb{E}(z_v|y_v = i)$ and $\mathbb{E}(z_v|y_v = j)$ is*

$$\begin{aligned}
\Delta_{i,j} &= |\mathbb{E}(z_v|y_v = i) - \mathbb{E}(z_v|y_v = j)| \\
&= |h \cdot (\mu_{ii} - \mu_{jj}) + \frac{1-h}{C-1}(\mu_{ij} - \mu_{ji}) + \frac{1-h}{C-1} \sum_{k=1, k \neq i,j}^{C} (\mu_{ik} - \mu_{jk})| \\
&= |\frac{hC-1}{C-1}(\mu_{ii} - \mu_{jj}) + \frac{1-h}{C-1}(\sum_{k=1}^{C}(\mu_{ik} - \mu_{jk}))|
\end{aligned} \tag{14}$$

*We firstly consider the situation of $h \geq \frac{1}{C}$. When $h \geq \frac{1}{C}$, we can infer the upper bound of $\Delta_{i,j}$ as:*

$$\begin{aligned}
\Delta_{i,j} &\leq \frac{hC-1}{C-1}|\mu_{ii} - \mu_{jj}| + \frac{1-h}{C-1} \sum_{k=1}^{C} |\mu_{ik} - \mu_{jk}| \\
&= \frac{hC}{C-1}(|\mu_{ii} - \mu_{jj}| - \frac{1}{C} \sum_{k=1}^{C} |\mu_{ik} - \mu_{jk}|) + \frac{1}{C-1}(\sum_{k=1}^{C} |\mu_{ik} - \mu_{jk}| - |\mu_{ii} - \mu_{jj}|),
\end{aligned} \tag{15}$$

*And the lower bound of $\Delta_{i,j}$ is:*

$$\begin{aligned}
\Delta_{i,j} &\geq \frac{hC-1}{C-1}|\mu_{ii} - \mu_{jj}| - \frac{1-h}{C-1} \sum_{k=1}^{C} |\mu_{ik} - \mu_{jk}| \\
&= \frac{hC}{C-1}(|\mu_{ii} - \mu_{jj}| + \frac{1}{C} \sum_{k=1}^{C} |\mu_{ik} - \mu_{jk}|) - \frac{1}{C-1}(\sum_{k=1}^{C} |\mu_{ik} - \mu_{jk}| + |\mu_{ii} - \mu_{jj}|),
\end{aligned} \tag{16}$$

*Thus, when $|\mu_{ii} - \mu_{jj}| > |\mu_{ik} - \mu_{jk}|, \forall k \in \{1, ...C\}$ and $h \geq \frac{1}{C}$, both the upper bound and lower bound of $\Delta_{i,j}$ will decrease with the decrease of $h$.*

*Next, we will show that lower $h$ under the condition of $h \geq \frac{1}{C}$ will lead to higher variance of aggregated nodes. According to Eq.(12), the variance of $\{z_v : y_v = i\}$ can be written as:*

$$Var(z_v|y_v = i) = Var(\sum_{u \in \mathcal{N}(v)} \frac{1}{|\mathcal{N}(v)|} x_u|y_v = i)$$

*According to the assumption 1, the neighbor features are conditional independent to each other given the label of the center node. And for each neighbor node $u \in \mathcal{N}(v)$, we have $P(y_u = y_v|y_v) = h$, $P(y_u = y|y_v) = \frac{1-h}{C-1}, \forall y \neq y_v$. Therefore, for neighbor node $u \in \mathcal{N}(v)$ of node $v$ whose label is $i$, its features follow a mixed distribution:*

$$\begin{aligned}
&P(x_u|y_v = i) \\
&= \sum_{k=1}^{C} P(y_u = k|y_v = i)P(x_u|y_u = k) \\
&= h \cdot N(\mu_{ii}, \sigma_{ii}) + \frac{1-h}{C-1} \sum_{k=1, k \neq i}^{C} N(\mu_{ik}, \sigma_{ik})
\end{aligned} \tag{17}$$

*Using the variance of mixture distribution, the variance of node $v$ in class $i$ can be derived as*

$$Var(z_v|y_v = i) = \frac{1}{d}Var(x_u|y_v = i)$$

$$= \frac{1}{d}(\mathbb{E}[Var(x_u|y_u, y_v = i)] + Var[\mathbb{E}(x_u|y_u, y_v = i)])$$

$$= \frac{1}{d}\Big(h\sigma_{ii}^2 + \frac{1-h}{C-1}\sum_{k=1,k\neq i}^{C}\sigma_{ik}^2 + h\mu_{ii}^2 + \frac{1-h}{C-1}\sum_{k=1,k\neq i}^{C}\mu_{ik}^2 - (h\mu_{ii} + \frac{1-h}{C-1}\sum_{k=1,k\neq i}^{C}\mu_{ik})^2\Big)$$

(18)

*Let $\bar{\mu}_i = \frac{1}{C}\sum_{k=1}^{C}\mu_{ik}$ and $\sigma_i^2 = \frac{1}{C}\sum_{k=1}^{C}(\mu_{ik} - \bar{\mu}_i)^2$. Then Eq.(18) can be rewritten as the following equation:*

$$Var(z_v|y_v = i)$$

$$= \frac{1}{d}\Big(\frac{hC-1}{C-1}\sigma_{ii}^2 + \frac{C-hC}{C-1}(\frac{1}{C}\sum_{k=1}^{C}\sigma_{ik}^2 + \sigma_i^2)$$

$$+ \frac{hC-1}{C-1}\mu_{ii}^2 + \frac{C-hC}{C-1}\bar{\mu}_i^2 - (h\mu_{ii} + \frac{1-h}{C-1}\sum_{k=1,k\neq i}^{C}\mu_{ik})^2\Big)$$

(19)

*As $h \geq \frac{1}{C}$, we can set $p = \frac{hC-1}{C-1}, 0 \leq p \leq 1$ and $\frac{C-hC}{C-1} = 1 - p$. For the last three terms of Eq.(19), we have:*

$$\frac{hC-1}{C-1}\mu_{ii}^2 + \frac{C-hC}{C-1}\bar{\mu}_i^2 - (h\mu_{ii} + \frac{1-h}{C-1}\sum_{k=1,k\neq i}^{C}\mu_{ik})^2$$

$$= p\mu_{ii}^2 + (1-p)\bar{\mu}_i^2 - (p\mu_{ii} + (1-p)\bar{\mu}_i)^2$$

$$= p(1-p)(\mu_{ii} - \bar{\mu}_i)^2 \geq 0$$

(20)

*Combining Eq.(19) and Eq.(20), we are able to get the lower bound of the variance as:*

$$Var(z_v|y_v = i)$$

$$\geq \frac{hC-1}{d(C-1)}\sigma_{ii}^2 + \frac{C-hC}{d(C-1)}(\frac{1}{C}\sum_{k=1}^{C}\sigma_{ik}^2 + \sigma_i^2)$$

$$= \frac{hC}{d(C-1)}(\sigma_{ii}^2 - \sigma_i^2 - \frac{1}{C}\sum_{k=1}^{C}\sigma_{ik}^2) + \frac{1}{d(C-1)}(C\sigma_i^2 + \sum_{k=1}^{C}\sigma_{ik}^2 - \sigma_{ii}^2)$$

(21)

*When $\sigma_i > \sigma_{ii}$, we know that with the decrease of $h$, the lower bound of $Var(z_v|y_v = i)$ will increase. Similarly, $Var(z_v|y_v = j)$ will also increase with a lower $h$. Combining with $|\mathbb{E}(z_v|y_v = i) - \mathbb{E}(z_v|y_v = j)|$ will decrease with the decrease of $h$, we can conclude that when $h \geq \frac{1}{C}$, the graph with lower $h$ will lead to less discrimative aggregate representations.*

*We then prove when $h < \frac{1}{C}$, the decreasing of $h$ will increase the discriminability of the aggregated representations by averaging. Specifically, with Eq.(14), we can infer that when $h < \frac{1}{C}$ the upper bound of $\Delta_{i,j}$ will be:*

$$\Delta_{i,j} \leq \frac{1-hC}{C-1}|\mu_{ii} - \mu_{jj}| + \frac{1-h}{C-1}\sum_{k=1}^{C}|\mu_{ik} - \mu_{jk}|$$

$$= \frac{-hC}{C-1}(|\mu_{ii} - \mu_{jj}| + \frac{1}{C}\sum_{k=1}^{C}|\mu_{ik} - \mu_{jk}|) + \frac{1}{C-1}(\sum_{k=1}^{C}|\mu_{ik} - \mu_{jk}| + |\mu_{ii} - \mu_{jj}|),$$

(22)

*And the lower bound of $\Delta_{i,j}$ is:*

$$\Delta_{i,j} \geq \frac{1-hC}{C-1}|\mu_{ii} - \mu_{jj}| - \frac{1-h}{C-1}\sum_{k=1}^{C}|\mu_{ik} - \mu_{jk}|$$

$$= \frac{-hC}{C-1}(|\mu_{ii} - \mu_{jj}| - \frac{1}{C}\sum_{k=1}^{C}|\mu_{ik} - \mu_{jk}|) - \frac{1}{C-1}(\sum_{k=1}^{C}|\mu_{ik} - \mu_{jk}| - |\mu_{ii} - \mu_{jj}|),$$

(23)

*Thus, when $h < \frac{1}{C}$ and $|\mu_{ii} - \mu_{jj}| > |\mu_{ik} - \mu_{jk}|, \forall k \in \{1, ...C\}$, both the upper bound and lower bound of $\Delta_{i,j}$ will increase with the decrease of $h$.*

*For the variance of aggregated representations when $h < \frac{1}{C}$, we can infer its folowing upper bound with Eq.(19):*

$$
\begin{aligned}
&Var(z_v|y_v = i) \\
\leq& \frac{hC - 1}{d(C-1)}\sigma_{ii}^2 + \frac{C - hC}{d(C-1)}(\frac{1}{C}\sum_{k=1}^{C}\sigma_{ik}^2 + \sigma_i^2) \\
=& \frac{hC}{d(C-1)}(\sigma_{ii}^2 - \sigma_i^2 - \frac{1}{C}\sum_{k=1}^{C}\sigma_{ik}^2) + \frac{1}{d(C-1)}(C\sigma_i^2 + \sum_{k=1}^{C}\sigma_{ik}^2 - \sigma_{ii}^2)
\end{aligned}
\tag{24}
$$

*According to the assumption that $\sigma_i > \sigma_{ii}$, we know that with the decrease of $h$ under the condition of $h < \frac{1}{C}$ the upper bound of the $Var(z_v|y_v = i)$ will decrease. We can have the same conlcusion for $Var(z_v|y_v = j)$. Combining the trend that when $h < \frac{1}{C}$ $|\mathbb{E}(z_v|y_v = i) - \mathbb{E}(z_v|y_v = j)|$ will increase with the decrease of $h$, we can conclude that when $h < \frac{1}{C}$, the graph with lower $h$ will have more discriminative aggregate representations.*

*When $h = \frac{1}{C}$, we can get*

$$
\Delta_{i,j} = \frac{1}{C}|\sum_{k=1}^{C}(\mu_{ik} - \mu_{jk})|,
\tag{25}
$$

$$
Var(z_v|y_v = i) \geq \frac{1}{d}(\frac{1}{C}\sum_{k=1}^{C}\sigma_{ik}^2 + \sigma_i^2)|,
\tag{26}
$$

*If $\sigma_i > \sqrt{d}|\mu_{ik} - \mu_{ik}|, \forall k \in \{1, \ldots, C\}$, we can get $Var(z_v|y_v = i) > \Delta_{i,j}^2$. So when $h = \frac{1}{C}$ and $\sigma_i > \sqrt{d}|\mu_{ik} - \mu_{ik}|, \forall k \in \{1, \ldots, C\}$, the representations after the averaging process will be non-discrimative.*

## D   Proof of Theorem 2

**Proof 2**  *In this proof, we also consider a center node $v$ in class $i$. And we focus on one dimension of the node feature and aggregated representation. Specifically, for each dimension, the label-wise aggregation can be written as:*

$$
a_{v,k} = \sum_{u \in \mathcal{N}_k(v)} \frac{1}{|\mathcal{N}_k(v)|}x_u,
\tag{27}
$$

*where $a_{v,k}$ denotes the aggregated feature of neighbors in class $k$. Since $u \in \mathcal{N}_k(v)$, we know node $u$'s features $x_u$ follows distribution as $x_u \sim N(\mu_{ik}, \sigma_{ik})$. The mean of $a_{v,k}$ in Eq.(27) is given as:*

$$
\mathbb{E}(a_{v,k}|y_v = i) = \mu_{ik}.
\tag{28}
$$

*Then the absolute difference between $\mathbb{E}(a_{v,k}|y_v = i)$ and $\mathbb{E}(a_{v,k}|y_v = j)$ will be:*

$$
\Delta_{i,j}^k = |\mathbb{E}(a_{v,k}|y_v = i) - \mathbb{E}(a_{v,k}|y_v = j)| = |\mu_{ik} - \mu_{jk}|.
\tag{29}
$$

*Given the assumption that the features are conditionally independent given the label of center node, the variance of $a_{v,k}$ can be written as:*

$$
Var(a_{v,k}|y_v = i) = \begin{cases} \frac{1}{dh}\sigma_{ik}^2 & \text{if } k = i\ ; \\ \frac{C-1}{d(1-h)}\sigma_{ik}^2 & \text{else}, \end{cases}
\tag{30}
$$

*In label-wise aggregation, we generally concatenate the $\{a_{v,k} : k \in \{1, \ldots C\}\}$ for further classification. Therefore, the lower bound of discriminability can be given by the representation of the class that are most discriminative, which can be formally written as:*

$$
k^* = \arg\max_{k} \frac{(\Delta_{i,j}^k)^2}{Var(a_{v,k}|y_v = i)}
\tag{31}
$$

When $h \geq \frac{1}{C}$, we can get:

$$\frac{(\Delta_{i,j}^{k^*})^2}{Var(a_{v,k^*}|y_v=i)} \geq \frac{dh|\mu_{ii} - \mu_{ji}|^2}{\sigma_{ii}^2} \geq \frac{d|\mu_{ii} - \mu_{ji}|^2}{C\sigma_{ii}^2} \tag{32}$$

As for $h \leq \frac{1}{C}$, let $k \neq i$ we can infer that:

$$\frac{(\Delta_{i,j}^{k^*})^2}{Var(a_{v,k^*}|y_v=i)} \geq \frac{d(1-h)|\mu_{ik} - \mu_{jk}|^2}{(C-1)\sigma_{ik}^2} \geq \frac{d|\mu_{ik} - \mu_{jk}|^2}{C\sigma_{ik}^2} \tag{33}$$

*Therefore, if the condition that $|\mu_{ik} - \mu_{jk}| > \sqrt{\frac{C}{d}}\sigma_{ik}, \forall k \in \{1, \dots, C\}$ is met, we can infer from Eq.(32) and Eq.(33) that $\frac{(\Delta_{i,j}^{k^*})^2}{Var(a_{v,k^*}|y_v=i)} > 1$ regardless the value of the homophily ratio $h$. This shows that label-wise aggregation can preserve the context and ensure the high discriminability regardless the homophily ratio.*

# E   Additional Details and Experiments on Generated Graphs

---

**Algorithm 2** Algorithm of Generating Graphs

---

**Input:** $\mathcal{G} = (\mathcal{V}, \mathcal{E}, \mathbf{X})$, $\mathcal{Y}_L$, target homophily ratio $h$, and target node degree $d$
**Output:** $\mathcal{G}' = (\mathcal{V}, \mathcal{E}', \mathbf{X})$
 1: Split the edges $\mathcal{E}$ into heterophilic edges $\mathcal{E}_n$ and homophilic edges $\mathcal{E}_s$.
 2: **if** $|\mathcal{E}_s| \geq hd|\mathcal{V}|$ **then**
 3:    Sample $hd|\mathcal{V}|$ edges from $\mathcal{E}_s$ to get $\mathcal{E}'_s$
 4: **else**
 5:    Obtain $hd|\mathcal{V}| - |\mathcal{E}_s|$ homophilic edges by randomly link nodes in the same class
 6:    Combine $\mathcal{E}_s$ with added homophilic edges to obtain $\mathcal{E}'_s$
 7: **end if**
 8: Randomly sample $d(1-h)|\mathcal{V}|$ edges from $\mathcal{E}_n$ as $\mathcal{E}'_n$
 9: Get $\mathcal{E}'$ with $\mathcal{E}' = \mathcal{E}'_n \cup \mathcal{E}'_s$

---

## E.1   Process of Graph Generation

To verify the conclusion in Theroem 1, we generate graphs with different homophily ratios and average degrees on the large-scale crocodile graph. Specifically, the average node degree of the target generated graphs is varied by $\{5, 10, 20\}$. For each node degree, we will sample the heterophilic edges, i.e., edges linking nodes in different classes, and homophilic edges, i.e., edges linking nodes in the same class from the original crocodile graph in different ratios to obtain realistic graphs with different heterophily levels. The homophily ratios of the generated graphs range from 0 to 0.9 with a step of 0.1. Since crocodile itself is a heterophilic graph that do not contain many homophilic edges, there could be no enough homophilic edges to obtain a graph with high homophily and node degrees. In this situation, we will randomly link nodes in the same class to get the required number of homophilic edges for graph generation. For the train/validation/test splits of generated graphs, they are the same as the original crocodile graph. The algorithm of the graph generation process can be found in Algorithm 2.

## E.2   More Experiments on Generated Graphs

To verify our theoretical analysis that label-wise aggregation can lead to distinguishable representations regardless the heterophily levels under mild conditions, we also compare LW-GCN with GCN and GAT on the generated graphs with different homophily ratios and average node degrees. The label-wise aggregation is conducted with the pseudo labels and provided ground-truth labels as it is described in Sec.4.2.2. Since we only focus on the label-wise graph convolution in the experiments, the model selection module is removed here. The other settings are the same as description in Appendix B.1. The average results of 10 splits are shown in Fig. 5. From this figure, we can observe that the performance of LW-GCN is much better than the GCN and GAT when the heterophily level is

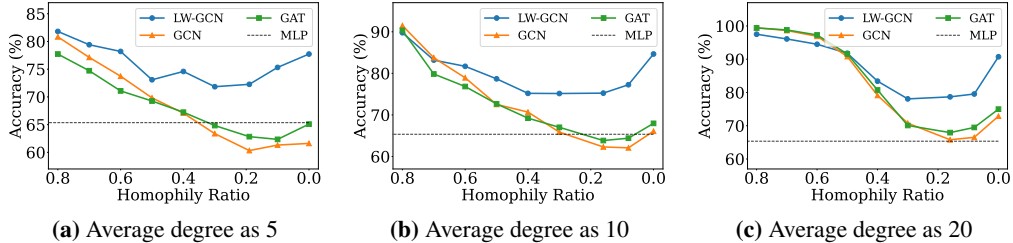

**Figure 5:** Comparisons between GCN, GAT and our LW-GCN on generated graphs. Note that model selection module is not adopted in LW-GCN in these experiments.

high. For example, when $h \approx 0.2$, both GCN and GAT can hardly outperforms MLP. By contrast, the accuracy of LW-GCN outperform GCN and GAT by around 10%. This demonstrates the effectiveness of adopting label-wise aggregation in graph convolution. In addition, we can find that only adopting the model with label-wise graph convolution will give slightly worse performance than GCN/GAT when the homophily ratio is very high. This implies the necessity of deploying a model selection module.

## F    Additional Experimental Results

The additional experimental results on Cornell and Citeseer datasets are presented in Table 4 and Table 5. The observations are similar to that of Table 1.

**Table 4:** Additional comparisons with GNNs originally designed for graph with homophily.

| Dataset | MLP | GCN | MixHop | SuperGAT | GCNII | LW-GCN |
|---|---|---|---|---|---|---|
| Cornell | 79.2 ±5.7 | 57.3 ±5.8 | 79.5 ±6.3 | 57.3 ±4.3 | 80.3 ±5.3 | **83.2 ±5.5** |
| Citeseer | 60.3 ±0.4 | 71.3 ±0.3 | 68.7 ±0.3 | 72.2 ±0.8 | 72.0 ±0.8 | **72.3 ±0.4** |

**Table 5:** Additional comparisons with GNNs designed for graph with heterophily.

| Dataset | FAGCN | SimP-GCN | H2GCN | GPRGNN | BM-GCN | ASGC | LINKX | GloGNN+ | LW-GCN |
|---|---|---|---|---|---|---|---|---|---|
| Cornell | 78.3 ±4.5 | 81.4 ±7.4 | 79.7 ±5.0 | 77.6 ±5.0 | 74.6 ±5.0 | 79.2 ±5.2 | 77.8 ±5.8 | **85.9 ±4.4** | 83.2 ±**5.5** |
| Citeseer | 71.7 ±0.6 | 71.8 ±0.8 | 71.0 ±0.5 | 71.1 ±0.9 | 68.9 ±1.0 | 70.2 ±0.2 | 51.6 ±1.7 | 66.7 ±1.9 | **72.3 ±0.4** |

## G    Impacts of Label-Wise Aggregation Layers

In this section, we explore the sensitivity of LW-GCN on the depth of $f_C$, i.e., the number of layers of label-wise message passing. Since LW-GCN will not select $f_C$ for homophilic graphs. We only conduct the sensitivity analysis on heterophilic graphs. We vary the depth of $f_C$ as $\{2, 3, \ldots, 6\}$. The other experimental settings are the same as that described in Sec. B.1. The results on Chameleon and Squirrel are shown in Fig. 6. From the figure, we find that our LW-GCN is insensitive to the number of layers, while the performance of GCN will drop with the increase of depth. This is because aggregation of LW-GCN is performed label-wisely to capture the context information. Embeddings of nodes in different classes will not be smoothed to similar values even after many iterations.

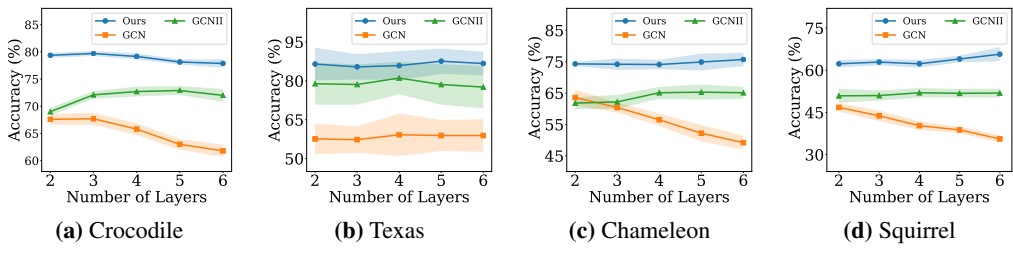

**Figure 6:** Classification accuracy with different model depth.

# H   Limitations of Our Work

In this paper, we conduct thoroughly theoretical and empirical analysis to show the impacts of heterophily levels to GCN. And we demonstrate the GCN model can be largely affected by heterophily and give poor prediction results. To alleviate the issue brought by heterophily, we develop a novel label-wise graph convolutional network to preserve the heterophilic context to facilitate the node classification. However, there are some limitations of our work. First, node labels are required for LW-GCN to obtain pseudo labels for label-wise graph convolution. However, in some tasks such as link prediction, labels are not available. Therefore, we will investigate how to obtain useful pseudo labels for applications that do not provide node labels. Second, in our theoretical analysis, we make several assumptions for simplification. Concretely, we conduct analysis on the d-regular graph. Following [50, 31], we also make an assumption on the label distribution of neighbor nodes. In our analysis, the node features are simplified to normal distribution and dimensions of features are independent to each other. These assumptions may not hold for some real-world graphs. For example, node degrees of the real network can be unbalanced which will contradict the assumption of d-regularity. The label distributions and feature distributions of neighbor nodes can be much more complex. Therefore, we will investigate the theoretical analysis on more flexible assumptions in the future. Third, recent studies [31, 35] show that the edge homophily ratio used in this paper could have significant drawbacks especially when the distribution of classes is unbalanced. To address these drawbacks, new measures such as adjusted homophily and label informativeness are proposed [35]. We leave the extension of our analysis on these new homophily measures as the future work.

