# OpenReview forum: "Label-Wise Graph Convolutional Network for Heterophilic Graphs"
_logconference.io/LOG/2022/Conference — LoG 2022 Poster_

### Official Review · Reviewer_nRvh · 2022-09-24

**Overall Score:** 6
**Confidence:** 4

**Review:**

**1. Summary of Contributions**

This paper considers the problem of learning on graphs with heterophily (and homophily) in which an edge tends to connect vertices belonging to different classes (and the same class respectively).

**The key insight** in the heterophilic setting is that, the neighbourhood sets of two different vertices, say v1 and v2, differ a lot if v1 and v2 belong to different classes, say c1 and c2, even when the said neighbourhood sets are restricted to contain vertices all of the same third class (say c3) but, crucially, the neighbourhoods are relatively more similar if c1=c2.

Using the aforementioned insight, the authors
1. predict classes (or labels) of unlabelled vertices initially using an MLP,
2. propose a label-wise graph neural network (GNN) which aggregates neighbourhood representations separately in a label-specific manner (and concatenates them),
3.  use another (homophily-based) GNN since it is not known, apriori, whether the graph exhibits homophily/heterophily,
4. combine the predictions of the two GNNs using bi-level optimisation, and
5. empirically demonstrate the effectiveness on homophilic and heterophilic datasets.



\
**2.Strong Points and Weak Points**

**Strong Points**

\+ The paper is well-organised and technically sound.

\+ Theoretical analysis (Theorems 1 and 2) and empirical evidence (Appendix F) are provided in support of the aforementioned key insight.

\+ The proposed model can learn effective node embeddings under both homophilic and heterophilic settings (Table 1) and better preserve the target nodes' features and neighborhood (Figure 3) with justifiable, ablated model components (Table 2).

**Weak Points**
\- The proposed method should be positioned and compared with existing relevant methods for heterophilic (and homophilic graphs). For example, one very relevant model [1] also predicts labels initially through an MLP and another work [2] uses node attributes as weak labels. Adaptive node-specific aggregation strategies [1, 3] have also been effective.
1. Powerful Graph Convolutional Networks with Adaptive Propagation Mechanism for Homophily and Heterophily, In AAAI'22
2. Diverse Message Passing for Attribute with Heterophily, In NeurIPS'21
3. Meta-Weight Graph Neural Network: Push the Limits Beyond Global Homophily, In TheWebConf'22

\- Both homophilic and heterophlic datasets considered in the paper are small and have issues (e,g., overfitting) as pointed out by recent publications on larger homophilic [4] and heterophilic  datasets [5]. Experiments on larger datasets will strengthen the significance of the contributions.

4. Open Graph Benchmark: Datasets for Machine Learning on Graphs, In NeurIPS'20,
5. Large Scale Learning on Non-Homophilous Graphs: New Benchmarks and Strong Simple Methods, In NeurIPS'21.


\
**3.Recommendation**

Weak accept.

While the key claims made in the paper are well-supported by theoretical and empirical analyses, the significance of the contributions with respect to prior work can be improved by positioning with relevant existing methods backed with experiments on larger datasets.


**4. Supporting Arguments for Recommendation**

The paper is of good quality with the key idea of the proposed method supported by solid theoretical (Theorems 1, 2) and empirical analyses (Appendix F). The components of the proposed method are also justified through an ablation study (Table 2). The main caveats are that the contributions need to be positioned with respect to relevant prior work [1, 2, 3] and experiments need to be conducted on larger datasets for the reason that the datasets used in this paper are small and known to have issues  (e,g., overfitting) [4, 5].

\
**5. Questions to Authors**
1. How do we adapt the method to graphs in which node features (or attributes) are not given as input?
2. How do we extend/adapt the proposed method when homophilic and heterophilic edges are given to us beforehand (e.g., signed networks)?
2. How do  we extend the proposed method for multi-relational edges? An example graph is a fraud-detector graph [H2-FDetector: A GNN-based Fraud Detector with Homophilic and Heterophilic Connections, In TheWebConf'22].


**6.Additional Questions/Feedback to Authors**

* Typo: The paper uses the word "discrimative" in many places, e.g., on lines 153, 176, 193, instead of discriminative.

* Have the authors tried comparing with a combination of a state-of-the-art heterophilic GNN and a homophilic GNN (predict $\phi_1, \phi_2$ using bi-level optimisation). This baseline does not need MLP to predict pseudo labels.

* How does the model compare with simplified graph convolution (SGC) which, for node classification, is essentially a logisitic regression-based model? A recent work proposes an adaptation of SGC to heterophilic datasets [Simplified Graph Convolution with Heterophily, In NeurIPS'22].

\
**7. Type of Paper**

9-page track

---

### Official Review · Reviewer_DvXN · 2022-10-19

**Overall Score:** 6
**Confidence:** 5

**Review:**

This paper proposes a label-wise graph convolutional network, specially designed for heterophilous graphs. To automatically select the GNN model for heterophilous/homogeneous graphs, the authors design a bi-level optimization objective. The paper also provides detailed theoretical analysis on the influence of heterophily on GNNs.

Strength:
1. The paper is very well written and easy to follow.

2. The theoretical analysis is clear and reasonable, which explains the reason why GCN could fail on heterophilous graphs.

3. The paper conducts extensive experiments to show the effectiveness of the proposed method.


Weakness:
1. The proposed method could heavily rely on the predicted pseudo labels of unlabeled objects. From the pseudocode of the proposed algorithm, f_P is only trained once. However, this could induce a larger number of incorrect predictions for unlabeled objects. So why not adopt a boostrapping strategy to include f_P in the loop and update the predicted pseudo-labels in each round? This is because when using MLP as the model, the training cost will not increase much.

2. I still have some doubts on the experimental results. In the ablation study part, from both Tables 2 and 4, we can see that the advantage of LW-GCN over LW-GCN\P is marginal on three graphs with heterophily: Crocodile, Chameleon and Squirrel. This is very strange because the latter leverages GCN to predict pseudo-labels, but GCN performs very poorly on these three datasets. So could you please explain this result?

3. For Cora, Citeseer and Pubmed, the used division in the experiments are not commonly used. Please use the datasets in [1] for fair comparison.
[1] Geom-gcn: Geometric graph convolutional networks

4. Some important baselines are missing:
[2] Finding Global Homophily in Graph Neural Networks When Meeting Heterophily
[3] GBK-GNN: Gated Bi-Kernel Graph Neural Networks for Modeling Both Homophily and Heterophily

#################################################################################

Overall recommendation: weak acc. Since the paper provides theoretical motivation for the design of label-wise graph convolutional networks and also extensive emprical experiments to verify the proposed model, the overall quality of the paper is above the acc line. So I recommend a weak acc.

Minor issues

Typos: line 155: inter-calss -> inter-class, line 193 resulting -> resulting in, line 319 one-hideen -> one-hidden, line 365 discrimative ->discriminative

---

### Official Review · Reviewer_esqv · 2022-10-20

**Overall Score:** 6
**Confidence:** 4

**Review:**

## Strengths

1. The assumption that features of heterophilic neighbors for nodes from different classes are different is interesting and insightful.
2. The writing is good.

## Weakness

1. Some missed comparisons and citations.


## Questions and Comments

1. Line 34-35, “we find the performance of GCN will firstly decrease then increase with the increment of  heterophily levels.” Missed citation. [1] found J-shaped curves and [2] found U-shaped curves under different levels of homophily.

2. “a heterophilic  context-preserving mechanism can lead to more discriminative representations.” This is similar to the claims in [3,4].

3. See a similar conclusion and empirical results as theorem 1 in [2].

4. Can the whole pipeline of LW-GCN be trained end-to-end or do we need to pretrain the MLP?

5.  What is the performance on Cornell and Film?

6. Comparison with some SOTA models, e.g. LINKX[4], GloGNN[6], ACM-GCN[2], BernNet[5].

Although there exist some problems, I think it is an interesting paper.

[1] Zhu J, Yan Y, Zhao L, et al. Beyond homophily in graph neural networks: Current limitations and effective designs[J]. Advances in Neural Information Processing Systems, 2020, 33: 7793-7804.

[2] Luan S, Hua C, Lu Q, et al. Is Heterophily A Real Nightmare For Graph Neural Networks To Do Node Classification?[J]. arXiv preprint arXiv:2109.05641, 2021.

[3] Ma Y, Liu X, Shah N, et al. Is homophily a necessity for graph neural networks?[J]. arXiv preprint arXiv:2106.06134, 2021.

[4] Chen J, Chen S, Huang Z, et al. Exploiting Neighbor Effect: Conv-Agnostic GNNs Framework for Graphs with Heterophily[J]. arXiv preprint arXiv:2203.11200, 2022.

[4] Lim D, Hohne F, Li X, et al. Large scale learning on non-homophilous graphs: New benchmarks and strong simple methods[J]. Advances in Neural Information Processing Systems, 2021, 34: 20887-20902

[5]  He M, Wei Z, Xu H. Bernnet: Learning arbitrary graph spectral filters via bernstein approximation[J]. Advances in Neural Information Processing Systems, 2021, 34: 14239-14251.

[6] Li X, Zhu R, Cheng Y, et al. Finding Global Homophily in Graph Neural Networks When Meeting Heterophily[J]. arXiv preprint arXiv:2205.07308, 2022.

---

### Official Review · Reviewer_dZBY · 2022-10-22

**Overall Score:** 5
**Confidence:** 4

**Review:**

**Summary:**

This paper proposes a label-wise message-passing approach for learning on heterophilous graphs. Namely, the idea is to separately aggregate the representations of neighbors belonging to different classes. For this, the neighbors' labels are needed, and the authors propose estimating them via MLP. It turns out that for homophilous graphs, the standard aggregation performs better, so it is suggested to train both models simultaneously and adaptively choose their mixing weights. Additionally, there is a theoretical analysis of the standard and new aggregation schemes.

**Strong points:**
- The proposed approach is reasonable and performs well on the considered datasets
- The idea of separately aggregating the neighbors from different classes is interesting and is supported by theoretical analysis

**Weak points:**
- The theoretical assumptions seem very strong and are not formal; see the detailed discussion below
- The experiments are conducted only on two types of datasets that have certain drawbacks (see details below)
- There are many typos and grammatical throughout the text

**Concerns regarding the theoretical analysis:**

I like that theoretical analysis illustrates the limitations of the existing aggregation and the usefulness of the proposed approach. However, the analysis seems to have very strong assumptions that can be hard to satisfy. General comment regarding the assumptions: to formally define them, a probability space should be introduced first. Otherwise, there can be ambiguities. The assumptions (Section 3.1) are the following:
1) d-regularity of a graph – real networks usually have heavy-tailed degree distributions, i.e., the degrees are highly unbalanced, contradicting this assumption.
2) neighbors’ features and class labels of a node are conditionally independent given the node’s label – this condition has to be formally defined, and I am not sure that I understand this correctly.
3) Given a node and its class, its neighbor has the same class with probability h (homophily ratio) and any other class with equal probability. This is an extremely strong assumption that prevents all non-trivial heterophily patterns. It essentially means that in the compatibility matrix, all non-diagonal elements are equal. This condition (together with d-regularity) also implies balanced class sizes.
4) Dimensions of node feature are independent – also seems to be not very realistic.
5) For a node in class i, its neighbors of class k should follow a normal distribution with parameters depending on both i and k. First, the assumption of a normal distribution seems to be strong. Second, the requirement of nodes in a class k having different distributions depending on their neighbors’ classes is also very strong. I am not sure that one can design a random graph model where this assumption is formally satisfied. Indeed, a node in class k may have neighbors of different classes. (Moreover, on average, it should have hd neighbors of the same class and d(1-h)/(C-1) neighbors of all other classes.)

In summary, many assumptions are quite strong, and I am not sure they can be theoretically simultaneously satisfied – an explanation or example is needed here.

In my opinion, this part requires an explicit statement about the limitations of the current assumptions. Alternatively, this part can be positioned as an intuitive explanation of the usefulness of the proposed idea.

Formulation of Theorem 1 - it is not clear what “discriminability” formally means (since Z is a matrix). Similarly, for Theorem 2, the statement “will keep its discriminability” is not formal.

**Concerns regarding the datasets:**

My main concern is the limited number of datasets. There are two types of datasets - Wiki and WebKB - giving five heterophilous datasets in total. The WebKB datasets are very small, while a recent paper [1] shows that the Wiki datasets have some duplicates, highly affecting the evaluation results. Thus, it can potentially be the case that the performance improvements on these datasets are caused by their somewhat degenerate structure.

**Some additional comments:**
- MLP is used to predict pseudo labels. This means that the approach will not work when there are no features and may work poorly if the features are not informative enough.
- As a measure of homophily, the paper uses so-called edge homophily. However, this measure was recently shown to have significant drawbacks, especially when class sizes are unbalanced [2,3]. This choice, however, does not seem significant for the current paper since the datasets considered in the experiments are not very unbalanced, and the theoretical analysis has strong assumptions that also seem to lead to balanced class sizes.
- “However, graphs are generally sparsely labeled which makes it difficult to estimate the real homophily ratio” – note that one may use the estimated pseudo labels here.

**Examples of typos / grammatical errors:**
- Line 32: missing space
- Line 33: redundant article before “Theorem”
- Line 34: missing space
- Line 70: “.” should be “;”
- Line 107: “to to”
- Line 127: “to” -> “into”
- Line 135: “i.e..” - redundant dot here
- Line 135: “each node have” – should be “has”
- Line 142: “mean.” - should be “,” here
- Line 145: “the molecule” – should be “a”
- Line 184: “lacking” -> “lack”
- Line 305: should be “?” instead of “.”
- Line 310: “the same as the cited papers” should probably be “in the cited papers”
- Line 332: “each heterophilic graphs” – should be “graph”
- Line 347: “the other GNNs that adopts” -> “adopt”
- Algorithm 1, line 2: “by with”
- Line 644: “Figure” -> “figure”
- Line 647: “This demonstrate” – should be “demonstrates”
- Line 689: “in same tasks” -> “in some tasks”

In summary, I do not advise accepting the paper in its current state

[1] A critical look at evaluation of GNNs under heterophily: Are we really making progress? https://openreview.net/pdf?id=tJbbQfw-5wv 2022

[2] Luan S. et al. Is Heterophily A Real Nightmare For Graph Neural Networks To Do Node Classification? arXiv:2109.05641. 2021.

[3] Platonov O. et al. Characterizing Graph Datasets for Node Classification: Beyond Homophily-Heterophily Dichotomy. ArXiv:2209.06177. 2022.

**Question to the authors:**

Q1: Clarifications regarding the concerns about the theoretical analysis (see above) would be very helpful.

Q2: Regarding the WebKB datasets – there are three standard datasets, two of which (Texas and Wisconsin) are considered in the paper. Why is the third one (Cornell) missing?

Q3: The results on Cora, Citeseer, and Pubmed seem weak compared to the GeomGCN paper (reference [23] in the current paper) – for instance, the performance of GCN/GAT in [23] is better than the reported best performance in Table 1. Is this caused by some difference in the setup?

Q4: The analysis in Section F is not clear. First, regarding the set X_{ik} defined in line 656 – may it have duplicates if node u has several neighbors of class i? Also, how exactly the average node feature cosine similarity between two sets is computed?

---

### Meta-Review · Area_Chair_4Fwv · 2022-11-11

**Confidence:** 5
**Recommendation:** Accept

**Meta Review:**

The paper explores the interesting case of graph with heterophilic features, a class of problems for which GNN typically struggle. A novel message passing algorithm is proposed to solve this problem and also studied theoretically. The authors show - under strong assumptions - that the proposed technique may lead to poor performances in the homophilic case and therefore propose a bi-level optimization method to select the right model. Reviewers broadly agree that these are novel contributions to a relevant problem in graph based learning and that the experiments proposed in the paper do confirm these findings. The paper is well written and clearly exposes the strengths and limitations of the method.

Originality, relevance, good mix of theory and experiments outweigh minor formatting issues and limited evaluations, I therefore recommend to accept the paper

---

### Decision · Program_Chairs · 2022-11-22

Accept (Poster)